# Insight into the Effects of Hydrodynamic Cavitation at Different Ionic Strengths on Physicochemical and Gel Properties of Myofibrillar Protein from Tilapia (*Oreochromis niloticus*)

**DOI:** 10.3390/foods13060851

**Published:** 2024-03-11

**Authors:** Kun Xie, Feng Yang, Xian’e Ren, Yongchun Huang, Fengyan Wei

**Affiliations:** 1Guangxi Key Laboratory of Green Processing of Sugar Resources, Key Laboratory for Processing of Sugar Resources of Guangxi Higher Education Institutes, School of Biological and Chemical Engineering, Guangxi University of Science and Technology, Liuzhou 565000, China; xiekun4649@126.com (K.X.); renxiane2014@126.com (X.R.); huangyc@yeah.net (Y.H.); weifengyan2022@163.com (F.W.); 2Guangxi Liuzhou Luosifen Research Center of Engineering Technology, Liuzhou 565000, China

**Keywords:** myofibrillar protein, hydrodynamic cavitation, ionic strength, physicochemical property, gelation property, microstructure

## Abstract

Effects of different ionic strengths (0.2, 0.4, and 0.6 mol/L) and different hydrodynamic cavitation (HC) treatment times (0, 1, 2, 3, and 4 min) on the conformation and gel properties of tilapia myofibrillar proteins (TMP) were investigated. The results showed that the solubility of TMP was significantly enhanced (*p* < 0.05) with the increase in NaCl concentration, and the gel characteristics were significantly improved. After HC treatment of TMP, the average particle size was significantly reduced (*p* < 0.05) and solubility was significantly enhanced (*p* < 0.05) with the increase in treatment time, the internal hydrophobic groups and reactive sulfhydryl groups were exposed. The intrinsic fluorescence spectra showed the unfolding of the spatial tertiary structure of proteins, and the circular dichroism spectroscopy showed the significant reduction in the content of α-helix in the secondary structure of the proteins (*p* < 0.05). In addition, the WHC and gel strength of the TMP heat-induced gels were enhanced, which improved the microstructure of the gels, and scanning electron microscopy showed that the gel network of the TMP gels became denser and more homogeneous. Dynamic rheology results showed that HC treatment resulted in a significant increase in the final G’ and G” values of TMP. In conclusion, HC treatment was able to improve the physicochemical structure and gel properties of TMP at different ionic strengths. This study presents a novel processing technique for the quality maintenance aspect of salt-reduced surimi gel products.

## 1. Introduction

As one of the freshwater economic fish, Tilapia is favored by the food industry as a result of its huge yield and high nutritional value [1]. Myofibrillar protein (MP) is a necessary component of tilapia protein [2,3], which is mainly composed of contractile proteins (myosin, actin) and regulatory proteins (tropomyosin, troponin) [4]. In addition to playing an important role in improving the quality characteristics such as stability and water-holding capacity (WHC) of surimi products (e.g., fish sausages, fish meatballs, etc.) [5,6], the thermal gelation properties of MP due to its thermal gelation effect also have a positive impact on the adhesion of surimi by aggregating meat product particles [7]. In recent years, protein gels have been extensively investigated by several scholars due to their potential applications in food and other fields [8]. Therefore, to enhance the value of meat products and produce gel-based healthy and nutritious meat products, investigating the physicochemical and gel properties of MP is of great significance.

In addition to being an essential ingredient in meat products to maintain good flavor, texture, and shelf life, NaCl is also an essential auxiliary in meat processing [9]. Being a salt-soluble protein, the functional properties of MP are influenced by many environmental factors including ionic strength [10]. Ionic strength has an effect on the solubility of MP and the interaction between proteins [11]. Wu et al. [12] observed a significant increase in the isoelectric point and neutral conditions of pork MP in solutions having NaCl concentrations of 0.2–0.8 mol/L with the tendency of aggregation between protein molecules decreasing. In addition, for thermal-induced gelation of MP, sufficient solubilization is a prerequisite and MP can form a good three-dimensional network structure gel only when the ionic strength reaches a certain level [13]. It was found that in the range of 0.2–0.6 mol/L, increasing ionic strength led to significant improvement in the gel strength, storage modulus (G’), WHC, and cooking yield [14,15,16] due to strengthening of molecular bonds between the heat-induced gels of MP. The above results show that the addition of a high concentration of NaCl has a positive effect on the formation of MP gel. However, long-term consumption of high-salt foods may lead to the development of diseases such as hypertension and increase the burden on the heart and kidneys. And the conditions of low ionic strength result in poor gelling properties of MP such as gel strength and WHC [17] resulting from difficult aggregation cross-linking between protein molecules due to poor solubility of salt-soluble MP. Therefore, low-sodium meat products are now a hot research area in the food industry [18].

To date, food researchers have tried to modify MP to improve its functional properties by using ultrasound, high pressure, and other physical techniques [19,20]. For example, Gao et al. [21] found that sonication was able to reduce cooking loss and increase the WHC of surimi gels by inducing a denser microstructure in silver carp MP gels under low-salt conditions. Shi et al. [22] found that high-pressure homogenization increased the solubility of clams MP while decreasing the particle size, turbidity, shear stress, apparent viscosity, and viscosity coefficient. Moreover, Liu et al. [23] found that the gel strength, WHC, chewiness, and hardness of heat-induced mud carp MP gel improved under ultra-high-pressure treatment. Recently, hydrodynamic cavitation (HC) as a new processing technology has received extensive attention in the application of protein modification due to its ability to produce cavitation similar to ultrasonic cavitation. During HC treatment, cavitation bubbles are generated when the fluid passes through the local compression area, and during the sudden increase in space and pressure, instantaneous local cavitation effects such as high temperature, high pressure, shear force, and free radicals are produced due to collapse of the cavitation bubbles [24]. In addition, compared with, HC has several advantages over ultrasonic cavitation like requirement of simple equipment, the ability to process large volumes, ease of control, energy-saving and environmentally friendly process, and more appropriateness for large-scale production enterprises [25]. Li et al. [26] observed that HC treatment of milk protein concentrate reduced the viscosity of protein milk and improved the rheological properties. Asaithambi et al. [27] found that HC treatment of egg white proteins improved the foaming, emulsifying, and gelling properties, and also reduced the viscosity of egg white proteins, increased the in vitro digestibility and improved their rheological properties due to the shear thinning behavior of HC. In their study, Hou et al. [28] observed an enhancement in the emulsifying capabilities of MP following HC processing, indicating a potential for HC technology to enhance the functionality of meat proteins. Nonetheless, the effect of varying ionic strengths on the physicochemical and gelation properties of MP during HC treatment remains unreported.

This study was carried out to analyze the impact of different treatment times (1, 2, 3, and 4 min) at different ionic strengths (0.2, 0.4, and 0.6 mol/L NaCl) on the physicochemical and gelation properties of tilapia (*Oreochromis niloticus*) myofibrillar protein (TMP) via HC treatment. Experimental measurements of TMP particle size, solubility, surface hydrophobicity (H_0_), reactive sulfhydryl content, structural changes of protein, and gel strength, WHC, rheological properties, and microstructure of TMP gels were carried out to elucidate the intrinsic mechanism of HC affecting the physicochemical and gelling properties of TMP at different ionic strengths.

## 2. Materials and Methods

### 2.1. Materials

Tilapia (800–1000 g/tail) was acquired from a nearby fresh produce market (Liuzhou, China) and was subsequently cut into portions and preserved in polyethylene bags at a temperature of −18 °C until required for use. High-purity (98%) bovine serum albumin (BSA) was sourced from Yuanye Biotechnology (Shanghai, China). 5,5-dithiobis-(2-nitrobenzoic acid) (DTNB) was supplied by Aladdin Biotechnology (Ontario, CA, USA). A 2.5% glutaraldehyde fixative solution was purchased from Xilong Scientific (Shantou, China). All other reagents used in the study were of analytical quality.

### 2.2. Extraction of TMP

Extraction of TMP was carried out as per the details given in a previous study [28]. Firstly, tilapia fillets were finely chopped in a meat grinder using an ice-chilled separation buffer (0.1 M NaCl, 25 mM KCl, 3 mM MgCl_2_, and 4 mM EDTA-2Na) at pH 7.0. This mixture was subsequently diluted to five times its original volume with more of the cold separation buffer. The resulting fish slurry underwent thorough homogenization with a homogenizer (Ultraturrax T25, IKA, Königswinter, Germany), set at 9000 rpm for 2 min, with a 30 s pause after each minute to prevent overheating. The mixture was passed through a layer of gauze to sift out larger particles such as fish skin and connective tissue. The supernatant was then centrifuged at 3220× *g* for 15 min at a temperature of 4 °C, using a refrigerated centrifuge to effectively separate the solid components from the liquid. The pellet, now free of larger debris, was washed with a cold 0.1 M NaCl solution and recentrifuged. This washing and centrifugation cycle was repeated twice to ensure purity. Finally, the samples were thoroughly washed with pH 7.0 PBS buffer, passed through three layers of gauze, and centrifuged one last time to yield a final precipitate that was TMP. The purified TMP was used within 24 h after storing at 4 °C. The biuret method using BSA as the standard [29] was used for the determination of protein concentration of TMP suspension with the help of an UV-Vis spectrophotometer.

### 2.3. HC Treatment

The HC treatment was carried out based on previous research with slight modifications [25]. A schematic diagram of the HC equipment is shown in Figure 1. As shown in the figure, the flow of TMP suspension was adjusted by V_1_, while the upstream inlet pressure, which is the pressure at both ends of the single-orifice plate cavitator (diameter of a single hole was 3 mm and the thickness was 20 mm) was adjusted by V_2_. The pressure was measured by P_1_ and P_2_, and flow of sampling port was controlled by V_3_. The TMP sample was moved from the storage tank to centrifugal pump through the single orifice plate and came back into the storage tank. During the flow of TMP sample through the single orifice plate, cavitation bubbles were created as pressure in the orifice tube was lower than the upstream inlet pressure. With a further decrease in pressure, the bubbles continued to grow. Subsequently, when the pressure was suddenly increased at the downstream outlet of single orifice plate, the bubbles burst and were accompanied by cavitation effects such as shear, turbulence, and free radicals.

TMP suspension with a concentration of 60 mg/mL was constituted by dissolving required amount of TMP in 0.2, 0.4, and 0.6 mol/L NaCl solution, and adjusted the pH to 6.0. The TMP suspensions were subjected to HC treatment at a pressure of 0.15 MPa using a single orifice plate for different time intervals of 0, 1, 2, 3, and 4 min. The treated TMP samples were stored at 4 °C for further experiments.

### 2.4. Physicochemical Properties

#### 2.4.1. Average Particle Size and Particle Distribution

The Zetasizer (Nano-ZS90, Malvern Instruments Ltd., Malvern, UK) was utilized to measure both the average particle size and its distribution for TMP samples at a concentration of 1 mg/mL.

#### 2.4.2. Myofibrillar Protein Solubility

Following the experimental guidelines outlined by Zhao et al. [30], the solubility of proteins was assessed. A concentrated TMP suspension was prepared and diluted to a concentration of 1 mg/mL. Subsequently, 8 mL of this diluted TMP sample was obtained and subjected to centrifugation at 10,000× *g* for a period of 10 min. The Biuret method was employed to quantify the protein content, which involved measuring the optical density of the TMP solution at 540 nm, both before and after the centrifugation process. The resulting protein solubility was then calculated using the formula presented as Equation (1):Solubility (%) = C_1_/C_2_ × 100,(1)
where C_1_ and C_2_ correspond to the protein content in the supernatant post-centrifugation and the overall protein content pre-centrifugation, respectively.

#### 2.4.3. Determination of Surface Hydrophobicity

The procedure for analyzing H_0_ followed an established protocol with some adaptations [31]. A 5 mg/mL TMP solution was formulated, blended with 1 mL of the aforementioned TMP solution with 200 μL of a BPB solution prepared at a concentration of 1 mg/mL, and stood at room temperature for 2 h. Afterward, the mixture was centrifuged at 6000× *g* for 15 min, 0.5 mL of the supernatant was extracted and diluted with 4.5 mL of NaCl solutions at concentrations of 0.2, 0.4, and 0.6 mol/L, respectively. The optical density of the diluted supernatant was determined at a wavelength of 595 nm. The degree of surface hydrophobicity in TMP is quantified by the extent of BPB binding, as delineated in the corresponding Equation (2):H_0_ (μg) = 200 μg × (A_1_ − A_2_)/A_1_,(2)
where, A_1_ and A_2_ correspond the absorbance value of blank control sample and TMP samples, respectively.

#### 2.4.4. Measurement of Reactive Sulfhydryl Group Content

The assay for the reactive sulfhydryl groups in TMP samples was conducted using a variation of the method described by Shi et al. [22]. A TMP suspension at a concentration of 1 mg/mL was created, and 4.0 mL of this suspension was combined with 50 μL of a 10 mmol/L DTNB solution. The mixture was then allowed to incubate for 20 min at ambient temperature, followed by the measurement of its absorbance at 412 nm. The content of reactive sulfhydryl groups was determined using the provided Equation (3):R-SH content (μmol/g) = (A_412_ × D)/(E_M_ − C_TMP_) × 10^6^,(3)
where, A_412_, D, E_M_ and C_TMP_ correspond the absorbance of test sample at 412 nm, dilution factor (in this experiment which is 1.02), molar extinction coefficient (1.36 × 10^4^ M^−1^ · cm^−1^) and TMP concentration (mg/mL), respectively.

### 2.5. Secondary and Tertiary Structure

#### 2.5.1. Intrinsic Emission Fluorescence Spectroscopy

The intrinsic emission fluorescence spectra were analyzed according to a previously published method with slight modifications [20]. The prepared TMP samples (0.2 mg/mL) were measured at 300–400 nm using a Cary Eclipse Fluorescence Spectrophotometer (G9800A, Agilent Technologies, Inc., Santa Clara, CA, USA).

#### 2.5.2. Circular Dichroism (CD) Spectroscopy

The CD spectrum for the diluted TMP sample, with a concentration of 0.1 mg/mL, was recorded with a CD spectropolarimeter model Chirascan, scanning the far-UV range from 200 to 260 nm. The results were expressed as the three-scan average molar ellipticity (θ) (deg·cm^2^/dmol) minus the baseline value. The ratios of the four secondary structure fits of TMP were also analyzed using the CDNN (version 2.1) software.

### 2.6. Gel Properties

#### 2.6.1. Gel Preparation

The procedure for creating TMP gels involved the following sequence: portions (40 mL) of a 60 mg/mL TMP solution were poured into glass vials measuring 45 mm in diameter and 50 mm in length. These vials were immersed in a water bath maintained at 45 °C for a duration of 15 min and subsequently shifted to a bath set at 80 °C for 30 min. After this, the vials were quickly chilled to ambient temperature using an ice pack and subsequently refrigerated at 4 °C for an entire night.

#### 2.6.2. Gel Strength

Once the TMP gels had reached equilibrium at ambient conditions for 60 min, the gel strength was determined by using a TA-XT Plus texture analyzer (Stable Micro System Ltd., Godalming, UK). The operation method is referred to Wang et al. [32]: during the compression test of the TMP gels, a P/0.5 cylindrical probe was utilized, with the test parameters adjusted to a compression distance of 5.0 mm, a speed of 2.0 mm/s, and a trigger force of 5.0 g.

#### 2.6.3. Gel Water-Holding Capacity

The WHC of TMP gels was measured by the centrifugal method according to Zhang et al. [33] with slightly modification. The gel samples of TMP, each weighing 5 g, were positioned within centrifuge tubes and subjected to centrifugation at 10,000× *g* for a period of 15 min at a temperature of 4 °C. After centrifugation, any excess water atop the gels was absorbed and removed with the aid of a filter paper. The WHC was expressed as Equation (4):WHC (%) = (m_2_ × m_0_)/(m_1_ − m_0_) × 100,(4)
where m_0_, m_1_ and m_2_ correspond the tube weight, the gel and tube weight before centrifugation, the gel and tube weight after centrifugation, respectively.

#### 2.6.4. Dynamic Rheological Measurements

The assessment of the rheological characteristics of TMP was conducted through temperature-dependent tests using a rotational rheometer model MCR72 from Anton Paar. Diluted TMP samples (40 mg/mL) were uniformly applied to the test platform and heated from 25 °C to 80 °C at a heating rate of 1 °C/min using a PP50 probe with a set spacing of 1.0 mm. The frequency and strain were set to 0.1 Hz and 2%, respectively. The storage modulus (G’) and loss modulus (G’’) were recorded as the test indexes for analysis.

#### 2.6.5. Scanning Electron Microscopy (SEM)

In preparation for SEM analysis, the TMP gel samples were immersed in a 2.5% glutaraldehyde solution for a period of 24 h. Subsequently, the samples were treated by removing the fixative and washing the gels three times using a 0.2 mol/L PBS solution. Then, a series of increasing ethanol concentrations were used for ethanol dehydration. The samples were freeze dried using a freeze dryer and sputter-coated with 10 nm of gold. The microstructure of TMP gel samples was observed using a scanning electron microscope (MIRA LMS, TESCAN, Brno, Czech Republic) at an accelerating voltage of 15 kV and a magnification of 1000×.

### 2.7. Statistical Analysis

Three replicate independent experiments were carried out to obtain the experimental data and expressed as mean ± standard deviation (SD). Significant differences (*p* < 0.05) between the different experimental groups were obtained by carrying out one-way analysis of variance (ANOVA) and Duncan’s test on the experimental data using SPSS (version 26.0, IBM Corp., Armonk, NY, USA) statistical software.

## 3. Results

### 3.1. Physicochemical Properties

#### 3.1.1. Average Particle Size and Particle Distribution

The functional properties of proteins is directly related to the particle size [34]. As shown in Figure 2a, for HC treatment time of 0–2 min, the average particle sizes of TMP decreased for all the ionic strengths studied (0.2, 0.4, and 0.6 mol/L) and each reached a minimum for the treatment time of 2 min. Compared with the control group (0 min), HC treatment for 1, 2, 3 and 4 min at ionic strengths of 0.2, 0.4 and 0.6 mol/L significantly reduced the average particle size of TMP (*p* < 0.05). From Figure 2b, it was observed that in the presence of 0.2, 0.4, and 0.6 mol/L ionic strengths, compared to the control group (0 min), particle size distributions of the TMP samples after HC treatment for different time intervals were bimodal and the peaks of particle distributions were shifted towards smaller sizes after HC treatment, which indicated fragmentation of the TMP aggregates into smaller particles due to the HC treatment. Such occurrences could potentially stem from the cavitation effects, including shear and turbulence, which occur during HC processing. These effects may disrupt the intermolecular non-covalent bonds (hydrogen and ionic bonds) among TMP molecules, resulting in the disassembly of TMP aggregates. Ren et al. [25] also found in their study that HC was able to reduce the particle size of soybean isolate proteins. 

However, as evident from Figure 2a, prolonged HC treatment time (3–4 min) resulted in a significant increase (*p* < 0.05) in the mean particle size of TMP. Similar results have been observed while studying Liu et al. [34] and Zhang et al. [35]. The reason might be related to the fact that the high temperature and free radical effect produced by prolonged HC treatment might be responsible for promoting the aggregation of TMP. Overall, appropriate HC treatment can effectively reduce the formation of TMP macro-aggregates even at low ionic strength.

#### 3.1.2. Myofibrillar Protein Solubility

Solubility of MP is closely related to the degree of aggregation of its molecules [12]. As shown in Figure 3a, when there was no HC treatment (0 min), increasing the ionic strength from 0.2 mol/L to 0.6 mol/L resulted in the increase in TMP solubility 5.43 ± 1.25% to 58.61 ± 3.11%. The increase in solubility can be attributed to how the addition of NaCl reduces the electrostatic interactions between protein molecules [2]. In addition, increasing HC treatment time (0–2 min) resulted in the increase in solubility of TMP at all the ionic strengths studied (*p* < 0.05). Decreasing particle size due to cavitation effects which increase the specific surface area and thus enhance the protein-water interaction might be responsible for the increase in TMP solubility [35]. Various scholars have also reported increased solubility of proteins as a result of HC treatment earlier [25,26,27].

#### 3.1.3. Surface Hydrophobicity

The H_0_ indicates the variation in the exposure of hydrophobic groups within the protein structure [34]. As shown in Figure 3b, the H_0_ values at an ionic strength of 0.2 mol/L were significantly lower than those at ionic strengths of 0.4 and 0.6 mol/L (*p* < 0.05). The addition of NaCl reduces the degree of aggregation of protein molecules and the exposure of hydrophobic groups increases [11]. Furthermore, the H_0_ values of TMP at all ionic strengths (0.2, 0.4, and 0.6 mol/L) increased significantly (*p* < 0.05) with increasing HC treatment time (0-2 min). Dissociation of TMP aggregates due to the shear stress, turbulence, and other cavitation effects caused by HC, resulting in full exposure of the hydrophobic groups to the outside which were able to bind more BPB, which might be responsible for the increase in H_0_ [22]. Conversely, the H_0_ of the TMP decreased significantly (*p* < 0.05) when the HC treatment time was extended to 3–4 min. Excessively prolonged HC treatments can lead to a decrease in H_0_ as a result of the transient high temperatures generated and free radical effects that lead to the re-polymerization of TMP. Zhao et al. [30] also found that high-pressure processing was able to unfold the pork MP, exposing the internal hydrophobic groups, but too much stress can lead to increased aggregation between proteins.

#### 3.1.4. Reactive Sulfhydryl Group Content

Reactive sulfhydryl groups situated on protein surfaces are indicative of alterations in protein conformation [36]. As shown in Figure 4a, at ionic strengths of 0.2, 0.4, and 0.6 mol/L, the active sulfhydryl content of TMP samples increased from 30.16 ± 0.55, 33.74 ± 0.34, and 35.84 ± 0.23 μmol/g to 35.09 ± 0.71, 38.78 ± 1.32, and 44.66 ± 0.24 μmol/g, respectively, following a 2 min of HC treatment, as compared to the control group (0 min). This might be due to the exposure of buried sulfhydryl groups in proteins during the breakdown of TMP aggregates due to the cavitation effect of HC treatment [31]. However, prolongation of HC treatment time resulted a substantial decrease in the content of reactive sulfhydryl groups (*p* < 0.05) for all the ionic strength values which could be attributed to the re-aggregation of TMP resulting from the transient high temperature generated by HC and subsequent encapsulation of the reactive sulfhydryl groups inside the proteins. On the other hand, as reported by Shi et al. [22], it might be due to the oxygen radicals generated by HC that induced the conversion of reactive sulfhydryl groups on the surface of TMP to disulfide bonds.

### 3.2. Secondary and Tertiary Structure

#### 3.2.1. Intrinsic Fluorescence Spectra

Typically, tryptophan (Trp) residues located in the internal hydrophobic environment when the protein is in the folded state have high fluorescence intensity and are particularly sensitive to the polarity of the surrounding microenvironment. Consequently, the intensity of fluorescence and the specific wavelengths (λ_max_) at which maximum fluorescence is emitted can serve as indicators for assessing the tertiary structure of proteins [31].

The fluorescence intensity of the TMP control sample (HC-0 min) increased with increasing ionic strength (Figure 4b). This implies that the introduction of NaCl facilitated the exposure of additional Trp residues within the protein, thereby enhancing its intrinsic fluorescence [11]. Furthermore, Figure 4b indicates that the peak fluorescence intensity of the TMP samples subjected to HC treatment progressively diminished as the duration of the treatment was extended from 0 to 3 min. Concurrently, there was an observed red-shift in the λmax of TMP, with a rise from 332 nm to 336 nm. The findings revealed that the mechanical forces, including shear and turbulence, produced during HC processing disrupted the protein’s hydrophobic interactions. This disruption caused the TMP molecular structure to unfold, allowing the internal Trp residues to be exposed to a more hydrophilic and polar environment, which in turn led to a decrease in the intensity of fluorescence [8]. Jiang et al. [20] reported that ultrasonication can enhance the polar environment through mechanical effects and can expose the tryptophan moiety to solvents, thus reducing the fluorescence intensity of MP. However, the fluorescence intensities of the TMP samples all increased when the HC treatment time reached 4 min, which could be attributed to the cavitation effect produced by HC inducing the refolding of the TMP molecules, which buried the exposed Trp residues [28]. These results indicate that proper HC treatment causes the unfolding of protein molecules, and the tertiary structure becomes loose, exposing internally buried hydrophobic groups with reactive sulfhydryl groups, which contributes to the formation of thermally induced gels by MP intermolecular cross-linking [30].

#### 3.2.2. CD Spectroscopy

The secondary structure of proteins can be assessed by CD spectroscopy [24]. In general, the secondary structure in MP is dominated by the α-helix conformation, and two negative peaks of α-helix can be seen at approximately 210 and 223 nm in the CD spectra, which suggests the presence of superhelical α-helix structure in the myosin tail [30]. Hydrogen bonding between the carbonyl oxygen (-CO) and amino hydrogen (NH-) of a polypeptide chain is mainly responsible for the formation of α-helix structure, which contributed to the stability of the secondary structure [8].

As shown in Figure 5, in the absence of HC treatment, increasing ionic strength resulted in an increase in absolute intensities of the two negative peaks, and Table 1 also shows the increasing α-helix content with increasing ionic strength. This indicated that the addition of NaCl promoted the formation of hydrogen bonds between proteins, increased the electrostatic repulsion, and increased the content of α-helix [37]. In addition, at ionic strengths of 0.4 and 0.6 mol/L, the intensity of both peaks of the CD spectroscopy decreased with increasing HC treatment time (0–4 min). And as can be seen from Table 1, the content of α-helix (*p* < 0.05) decreased significantly along with a significant increase in the content of β-sheet, β-turn, and random coil in all HC-treated groups (*p* < 0.05), suggesting that HC facilitates the unfolding of secondary structure by disrupting the hydrogen bonds within the TMP molecule [28]. Jiang et al. [20] also reported in their study that ultrasonic treatment led to a decrease in α-helix and β-turn content in MP.

On the contrary, for 0.2 mol/L ionic strength, increasing HC treatment time (0–4 min) resulted in an initial increase in the content of α-helix followed by a decreasing trend (Table 1). The increase in α-helix content may be attributed to the cavitation effect produced by HC which promotes the reconstruction of hydrogen bonds within the protein molecule and stabilizes the hydrogen bonds within the protein peptide chain. However, the decrease in α-helix content with HC treatment time exhibited a molecular conformational change that transformed the α-helix to β-folding, a change similarly found in the study by Asaithambi et al. [27].

### 3.3. Gel Properties

#### 3.3.1. Gel Strength

Gel strength is an important parameter of MP performance and is closely related to the sensory quality of meat products [17]. Figure 6a demonstrates the variation in gel strength of TMP with varying HC treatment times in the presence of different ionic strengths. Increasing NaCl concentration from 0.2 mol/L to 0.6 mol/L resulted in a significant increase (*p* < 0.05) in the gel strength of TMP control (0 min) samples. This could be ascribed to the impact of NaCl on the degree of TMP swelling, which leads to a reduction in protein aggregation and an increase in solubility [2]. At low salt concentrations (0.2 mol/L), low gel yield results from poor protein solubility due to the existence of MP mainly in a filamentous state [16]. In addition, increasing HC treatment times (0–2 min) significantly enhanced (*p* < 0.05) the gel strength of TMP gel samples at all the ionic strength values. This might be attributed to the reduction in TMP particle size due to the cavitation effect of HC, the formation of aggregates when heated for gelation due to cross-linking of the exposed hydrophobic groups, and the formation of disulfide bonds via the reaction of sulfhydryl groups, which promoted cross-linking of the proteins and contributed to the enhancement of the gel network [36]. In a similar report, Zhang et al. [33] found an improvement in the gel strength of MP via ultrasonic treatment. With further progress of the HC treatment time (3–4 min), a gradual lowering of gel strength was observed which may be attributed to the protein reaggregation induced by the high temperature generated by HC treatment, resulting in the reduction in the interaction force between the proteins. On the other hand, irregular aggregation of proteins during the heat-induced gelation and the formation of heterogeneous gel microstructures resulted in the reduction in the gel strength of the TMP gel [23].

#### 3.3.2. Gel Water-Holding Capacity

The gel network formed by protein–protein cross-linking constitutes the protein gels and WHC that can indicate the ability of protein gels to bind water, which is an important feature of heat-induced gels and an important factor in the assessment of the quality of meat products [7]. Figure 6b demonstrates the effect of different ionic strengths and different HC treatment times on the WHC of TMP gels. The WHC of the TMP gel control samples (0 min) gradually increased from 19.08 ± 2.78% to 44.24 ± 1.57% when the ionic strength was increased from 0.2 mol/L to 0.6 mol/L. This may be due to the fact that the increase in ionic strength promotes the formation of more hydrogen bonds between protein molecules and enhances the stability of proteins. Moreover, MP generally existed as fibrous filaments at low ionic strength (0.2 mol/L), and the thermally induced gel formed was porous, inhomogeneous, and had a large pore size, whereas the network of thermally induced gel formed by heating at high ionic strength (0.4 and 0.6 mol/L) had small pore sizes and uniform distribution and was capable of storing more water [14]. In addition, increasing HC treatment time (0–2 min) significantly increased (*p* < 0.05) the WHC of TMP gel samples in comparison to the TMP control (0 min). This could be because HC processing facilitates the structural expansion of proteins, thereby revealing a greater number of hydrophobic groups (Figure 3b) and reactive sulfhydryl groups (Figure 4a). This promotes the formation of additional disulfide and hydrogen bonds among TMP molecules as the thermally-induced gel sets. Consequently, this results in a more compact and consistent TMP gel structure upon heating, thereby enhancing its WHC [23]. However, with a further increase in HC treatment time (3–4 min), the WHC also decreased, showing the same trend as that of particle size, H_0_, and gel strength. The reason might be the presence of a large cavity structure in the gel network resulting from the formation of inhomogeneous TMP heat-induced gels due to the aggregation of protein molecules induced by the cavitation effect of HC thus lowering the WHC [31]. Zhang et al. [35] also pointed out in their study that the change in WHC might be related to the chemical force between proteins and the structure of the gel network formed by proteins.

#### 3.3.3. Dynamic Rheological Measurements

Dynamic rheological measurements are able to detect the formation process of thermally induced MP gel matrix, which is an important indicator to characterize the functional properties of MP. The measurement of storage modulus (G’) represents stored energy of the elastic part and the measurement of loss modulus (G”) represents energy dissipated as heat in the viscous part [35]. In general, the typical rheological pattern of MP can be divided into three processes: solidification (25–48 °C) → gel weakening (48–52 °C) → gel enhancement (52–80 °C). This is a process that reveals the formation of thermally induced gels of MP, as widely reported in previous studies [38]. The G’ and G” of TMP samples with different ionic strengths and after different HC treatment times during gelation are displayed in Figure 7. Increasing ionic strength from 0.2 mol/L to 0.6 mol/L showed a significant increase in the final values of G’ and G” of the TMP control samples (0 min), with the G’ values of 0.6 mol/L TMP samples being much higher than those of 0.2 mol/L. This phenomenon could be attributed to factors including the reduction in protein aggregation of TMP due to increased ionic strength, enhanced protein solubility, and the resulting elevation in the strength of the formed gel, subsequently leading to an increase in the G’ value [10]. In addition, with increasing time of HC treatment (0–2 min), G’ values of all the TMP samples increased to reach a maximum value at 2 min, indicating higher elasticity and stronger structure of the gel network formed by the TMP samples. HC treatment broke the aggregation of proteins which decreased the average particle size and exposed the hydrophobic groups and reactive SH groups, leading to an increase in the number of disulfide bonds during thermally induced gelation, and a subsequent increase in the final G’ values of TMP samples [19]. However, further increase in HC treatment time (3–4 min) resulted in the decrease in final values of both G’ and G”. This can be attributed to the excessive aggregation of proteins prior to the heating gelation process as a result of HC treatment, leading to a weakening of the strength of the formed TMP thermally induced gels, which is reflected in the same trend in the results for particle size (Figure 2a) and surface hydrophobicity (Figure 3b). Zhao et al. [30] also found in their study that the final values of G’ and G” of MPs were reduced by stronger (300 and 400 MPa) high-pressure processing.

#### 3.3.4. Gel Microstructure

SEM can directly show the microstructure of MP gels and reflect the changing gel properties [31]. The 1000× magnified SEM images of TMP gel samples in the presence of different ionic strengths and for different HC treatment times are shown in Figure 8 which showed significant differences in their microstructures. The TMP gel in the presence of ionic strength of 0.2 mol/L showed clustered cross-linked loosely structured gel filaments with several large cavities (Figure 8a), which explains lower gel strength (Figure 6a) and WHC (Figure 6b) of the thermally induced gel of TMP at 0.2 mol/L. With the increase in NaCl concentration (0.4 mol/L, and 0.6 mol/L), the structure of the TMP heat-induced gel changed into a three-dimensional network structure composed of cross-linked proteins with more homogeneous small cavities (Figure 8f,k), which resulted in improved gel strength and WHC of the TMP heat-induced gel. Feng et al. [10] found that the addition of NaCl was able to stabilize and tighten the three-dimensional network structure of MP gels, which was consistent with our results. It is also interesting to note that HC treatment resulted in improved microstructures of the TMP thermally induced gels, with a decrease in the pore size of the network structure and denser gel morphology. This phenomenon can be attributed to the effect of cavitation generated during HC treatment on the interactions between proteins and their spatial structure, such as the reduction in particle size (Figure 2), the exposure of more hydrophobic groups (Figure 3b), and reactive sulfhydryl groups (Figure 4a), and the weakening of intrinsic fluorescence spectra (Figure 4b), which are alterations contributing to the formation of TMP thermally-induced gels with a denser and more homogeneous three-dimensional network structure [4]. Zhao et al. [30] found that under high-intensity ultrasound, the SEM results show that the three-dimensional network structure of MP is improved. Overall, the TMP gel samples treated with HC for 2 min in the presence of an ionic strength of 0.6 mol/L exhibited the best microstructure (Figure 8m). However, compared with 0.6 mol/L TMP gel without HC treatment, the 0.4 mol/L TMP gel sample was able to form a more homogeneous three-dimensional network microstructure after 2 min of HC treatment (Figure 8h), which demonstrated that the HC technique has a great application in the development of low-sodium salt gel meat products. However, with further increase in the HC treatment time (3–4 min), the network structure of TMP heat-induced gels collapsed, and the three-dimensional network also began to shift towards a loose morphology, with the gel network structure becoming inhomogeneous and large pore-like structures started to appear. This is due to the transient high temperatures and reactive oxygen radicals generated by the prolonged HC treatment oxidizing the TMP, resulting in the proteins repolymerising in a disordered manner and forming a loose and irregular network of thermally induced gels [33].

## 4. Discussion

In this study, the results showed that the elevated ionic strength resulted in different degrees of improvement in both the physicochemical and gel properties of TMP. When HC has been treated for 2 min, a significant decrease (*p* < 0.05) in average particle size and a significant increase (*p* < 0.05) in solubility were observed, internal hydrophobic groups and reactive sulfhydryl groups were exposed, the α-helix in the secondary structure of TMP was reduced, the tertiary spatial structure was unfolded, the gel strength and WHC of the TMP heat-induced gel were significantly enhanced *(p* < 0.05), both G’ and G” were substantially enhanced, and SEM observations showed that TMP formed a compact and homogeneous gel structure. Therefore, the HC treatment at 2 min is the optimal treatment time to help improve the gel characteristics of TMP. Collectively, our study found that HC processing can improve the quality of MP gel properties by enhancing the physicochemical of MP, when salt levels are reduced. This improvement in texture and stability is particularly important for the food industry, as it meets consumer demand for healthier food options and innovative processing methods. The research has implications for a wider array of meat protein products, fostering innovation and variety in the market. However, additional research is essential to fully understand the complex effects of HC processing on fish proteins. Insight into these effects is vital for optimal the HC parameters to maintain the quality and uniformity of the end product. This study of HC treatment will not only advance the field of food science but also support the broader objectives of enhancing health and sustainability within our food supply.

## Figures and Tables

**Figure 1 foods-13-00851-f001:**
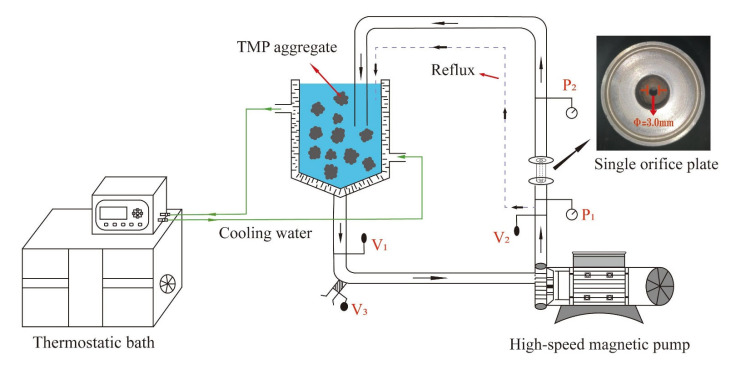
Explanation of the design concept behind HC apparatus.

**Figure 2 foods-13-00851-f002:**
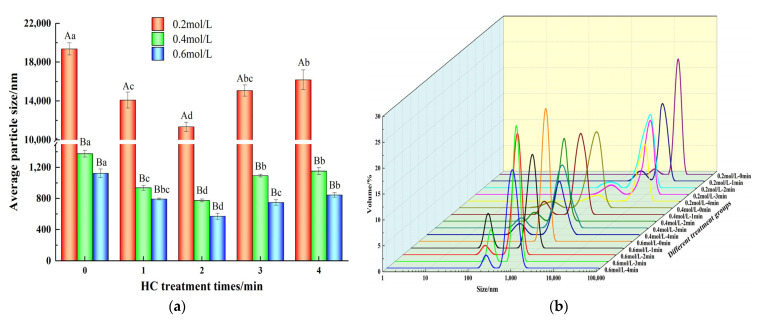
Effect of HC treatment times on the average particle size (**a**) and particle size distribution (**b**) of TMP under different ionic strengths. Different uppercase letters indicate that TMP showed significant differences at various ionic strengths (*p* < 0.05). Similarly, different lowercase letters showed significant differences in TMP at different HC treatment times (*p* < 0.05).

**Figure 3 foods-13-00851-f003:**
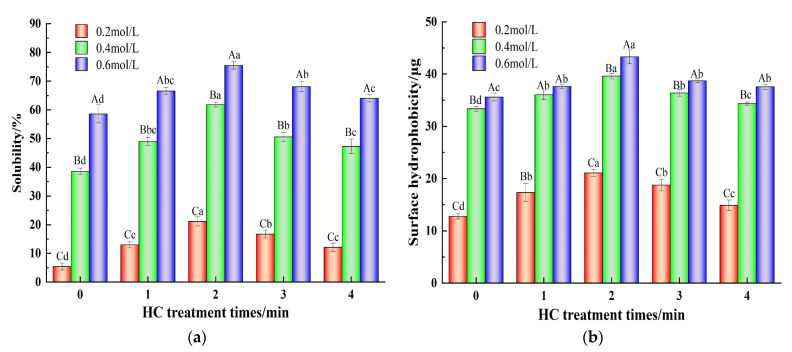
Effect of HC treatment times on the solubility (**a**) and surface hydrophobicity (**b**) of TMP under different ionic strengths. Different uppercase letters indicate that TMP showed significant differences at various ionic strengths (*p* < 0.05). Similarly, different lowercase letters showed significant differences in TMP at different HC treatment times (*p* < 0.05).

**Figure 4 foods-13-00851-f004:**
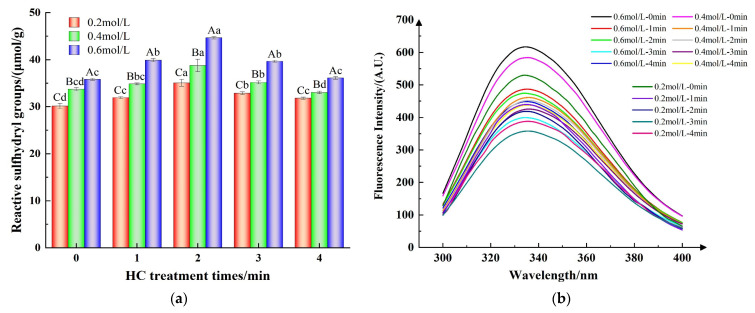
Effect of HC treatment times on the content of reactive sulfhydryl group (**a**) and intrinsic fluorescence spectra (**b**) of TMP under different ionic strengths. Different uppercase letters indicate that TMP showed significant differences at various ionic strengths (*p* < 0.05). Similarly, different lowercase letters showed significant differences in TMP at different HC treatment times (*p* < 0.05).

**Figure 5 foods-13-00851-f005:**
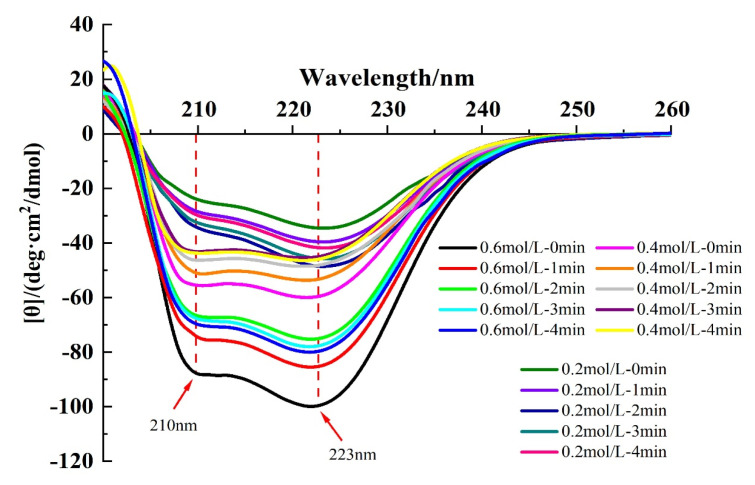
Effect of HC treatment times on the content of circular dichroism (CD) spectroscopy of TMP under different ionic strengths.

**Figure 6 foods-13-00851-f006:**
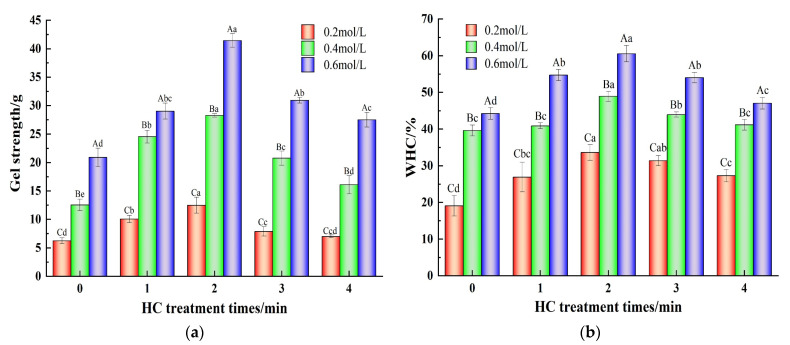
Effect of HC treatment times on the content of gel strength (**a**) and gel water-holding capacity (**b**) of TMP under different ionic strengths. Different uppercase letters indicate that TMP showed significant differences at various ionic strengths (*p* < 0.05). Similarly, different lowercase letters showed significant differences in TMP at different HC treatment times (*p* < 0.05).

**Figure 7 foods-13-00851-f007:**
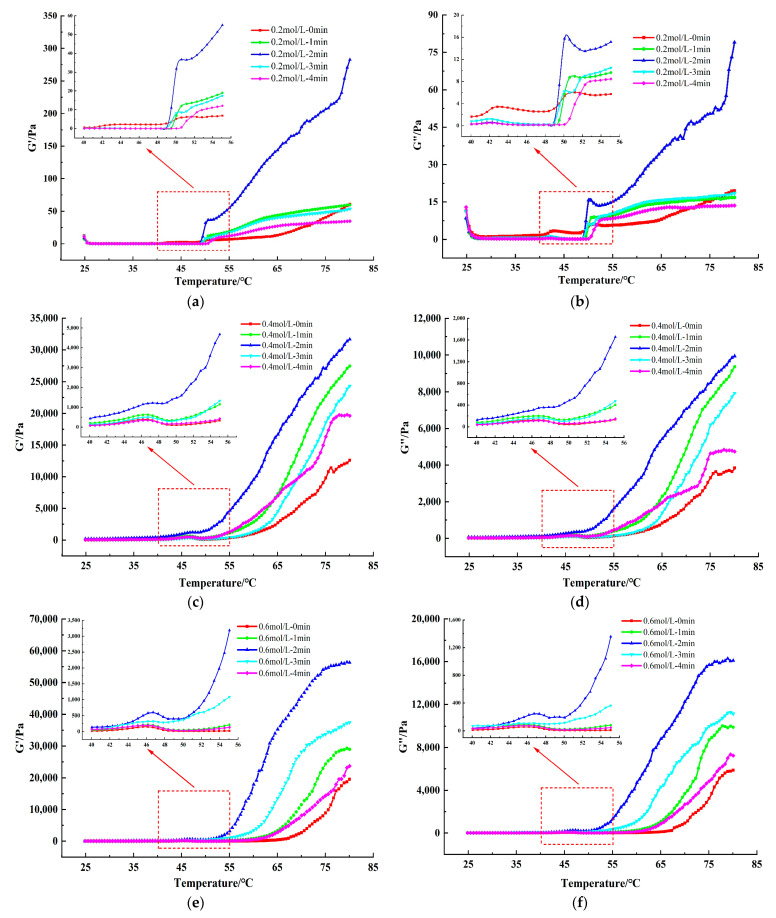
Effect of HC treatment times on the dynamic rheological storage modulus (G’) (**a**,**c**,**e**) and loss modulus (G’’) (**b**,**d**,**f**) of TMP under different ionic strengths. The inset figures show an enlarged view of the dynamic rheology at temperatures of 40–55 °C.

**Figure 8 foods-13-00851-f008:**
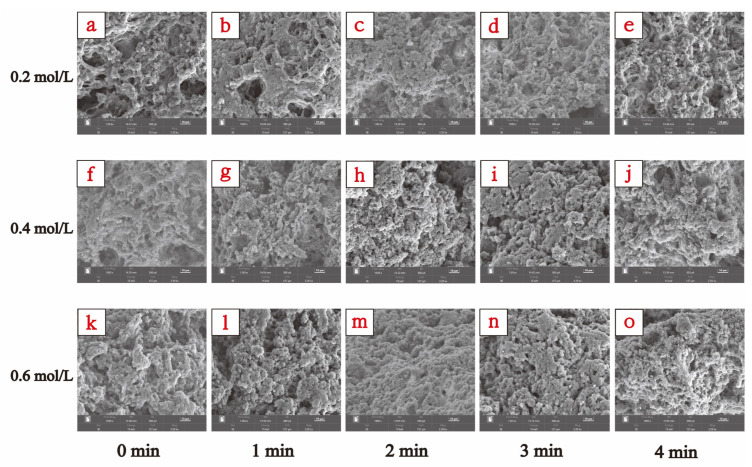
Effect of HC treatment times on the microstructure of TMP heat-induced gels in the presence of different ionic strengths. (**a**–**e**), (**f**–**j**), and (**k**–**o**) were the images observed by scanning electron microscopy (SEM) (1000× magnification) of induced TMP heat-induced gels at ionic strengths of 0.2, 0.4, and 0.6 mol/L for different HC treatment times (0, 1, 2, 3, and 4 min), respectively.

**Table 1 foods-13-00851-t001:** Effect of HC treatment times on the secondary structure content (%) of TMP under different ionic strengths.

Ionic Strength (mol/L)	HC Treatment Time (Min)	Secondary Structure Content of TMP(%) *			
		α-helix	β-sheet	β-turn	Random Coil
0.2	0	28.0 ± 0.4 ^Ce^	20.5 ± 0.7 ^Aa^	16.3 ± 0.1 ^Aab^	35.2 ± 0.4 ^Aa^
	1	34.4 ± 0.3 ^Bd^	16.0 ± 0.2 ^Ab^	16.6 ± 0.3 ^Aa^	33.0 ± 0.3 ^Ab^
	2	39.2 ± 0.2 ^Aa^	14.2 ± 0.1 ^Bc^	15.4 ± 0.1 ^Cd^	31.2 ± 0.1 ^Cd^
	3	37.7 ± 0.3 ^Ab^	15.3 ± 0.2 ^Bb^	15.3 ± 0.1 ^Cd^	31.7 ± 0.3 ^Bc^
	4	35.1 ± 0.3 ^Ac^	15.7 ± 0.2 ^Bb^	16.1 ± 0.3 ^Bc^	33.1 ± 0.3 ^Ab^
0.4	0	39.5 ± 0.8 ^Ba^	13.9 ± 0.2 ^Bb^	15.7 ± 0.4 ^Bc^	30.9 ± 0.2 ^Bb^
	1	35.8 ± 1.6 ^Bb^	15.8 ± 0.7 ^Aa^	16.1 ± 0.3 ^Bbc^	32.3 ± 0.4 ^Bab^
	2	34.9 ± 0.5 ^Bbc^	16.4 ± 0.4 ^Aa^	16.0 ± 0.1 ^Bc^	32.7 ± 0.4 ^Ba^
	3	33.7 ± 0.3 ^Cc^	16.3 ± 0.2 ^Aa^	16.5 ± 0.1 ^Aa^	33.5 ± 0.2 ^Aa^
	4	33.7 ± 0.2 ^Cc^	16.6 ± 0.1 ^Aa^	16.2 ± 0.2 ^Aa^	33.5 ± 0.3 ^Aa^
0.6	0	51.2 ± 0.2 ^Aa^	11.3 ± 0.2 ^Cd^	13.9 ± 0.1 ^Cd^	23.6 ± 0.3 ^Cc^
	1	39.3 ± 0.2 ^Ab^	13.6 ± 0.2 ^Bc^	15.4 ± 0.2 ^Cc^	31.7 ± 0.5 ^Bb^
	2	33.7 ± 0.2 ^Cc^	16.6 ± 0.2 ^Aa^	16.4 ± 0.2 ^Aa^	33.3 ± 0.2 ^Aa^
	3	34.3 ± 0.3 ^Bc^	16.0 ± 0.1 ^Ab^	16.1 ± 0.1 ^Bab^	33.6 ± 0.2 ^Aa^
	4	34.5 ± 0.3 ^Bc^	16.5 ± 0.4 ^Aa^	15.9 ± 0.3 ^Bb^	33.1 ± 0.1 ^Aa^

* Note: Data were presented as mean ± standard deviation (*n* = 3). Different uppercase letters indicate that TMP showed significant differences at various ionic strengths (*p* < 0.05). Similarly, different lowercase letters showed significant differences in TMP at different HC treatment times (*p* < 0.05).

## Data Availability

The original contributions presented in the study are included in the article, further inquiries can be directed to the corresponding author.

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
