# Peer review of "Insight into the Effects of Hydrodynamic Cavitation at Different Ionic Strengths on Physicochemical and Gel Properties of Myofibrillar Protein from Tilapia (Oreochromis niloticus)"

_foods, 2024, doi:10.3390/foods13060851_

Round 1

Reviewer 1 Report

Comments and Suggestions for Authors

The research is of interest and does cover an important area for investigation. One of the main aspects of understanding the role of protein structure surrounds the extraction method, so could you please highlight the extraction method used rather than just referencing a previous publication. This would aid an understanding of the factors involved in the research.

It is interesting to note the effect of HC on aggregation. Can you provide a little more evidence to support this claim, and can you discuss how previous research has illustrated this point in particular ?

Has other research on similar meat proteins been conducted and shown similar effects in protein solubilisation especially with regards to ionic strength ? How would this affect the bioavailability and digestibility of the proteins as well as their gelling properties?

The graphs and figures are analysed well and the statistical differences illustrated. However I was wondering if there was a chance to have a consistent brightness or contrast on the SEM images as this would improve comparisons.

The discussion section is quite simple and could be expanded for another paragraph or two to illustrate the novelty and the impact of the research. 

The introduction could also be improved with reference to recent research in similar protein from other meat sources.

Comments on the Quality of English Language

Generally written well and no real issues

Reviewer 2 Report

Comments and Suggestions for Authors

This study investigated the effect of different HC time at different ionic strengths on the physicochemical and gelation properties of tilapia myofibrillar protein for the maintaining quality of salt-reduced surimi product. I didn’t find any errors in experimental methods, results and discussion.

However, I am curious about the pH of the myofibrillar protein suspension before heating gelation. It is difficult for myofibrillar protein to form heating gel under pH 7.0. As you know, Myofibrillar protein is converted to actomyosin in high salt and pH 7 or higher and forms a cross-linking structure when heated to become a viscoelastic gel.

Line 159 : In equation, where did 200 ug come from?

Line188-189: FAO/WHO Food standard program recommend high temperature setting ability: 40°C for 30 min, followed immediately by 90°C for 15 min in time-temperature relationships for thermal processing (Park, J.W. 2005. Surimi and surimi seafood, Taylor & Francis, Boca Raton, U.S.A., p.877). What are the criteria for the heating process for gel.  
